# Cut-Out Towne-View Whole-Brain 320-Row Four-Dimensional Computed Tomography Angiography for Assessing the Anterior Intracranial Collateral Status: A Retrospective Study

**DOI:** 10.3390/diagnostics12061336

**Published:** 2022-05-27

**Authors:** Takahisa Mori, Toshimitsu Shimizu, Hirobumi Sato, Natsuki Mashikawa

**Affiliations:** 1Department of Stroke Treatment, Shonan Kamakura General Hospital, 1370-1, Kamakura City 247-8533, Kanagawa, Japan; 2Radiology Division, Shonan Kamakura General Hospital, 1370-1, Kamakura City 247-8533, Kanagawa, Japan; toshimitsu19792000@icloud.com (T.S.); hirobumi_sato1013@yahoo.co.jp (H.S.); na.teama181120@gmail.com (N.M.)

**Keywords:** computed tomography angiography, four-dimensional computed tomography, ischemic stroke, reliability, thrombectomy

## Abstract

Whole-brain four-dimensional computed tomography angiography (W4D-CTA) using a 320-row area detector CT (320r-ADCT) has been applied before thrombectomy. Endovascular physicians require images with high interrater reliability (IRR) for making appropriate decisions. However, the 320r-ADCT gantry cannot be tilted, and the patient’s head position influences the anteroposterior (AP)-view W4D-CTA images. This study aimed to determine which W4D-CTA images are appropriate pre-thrombectomy, whether the unedited AP view or cut-out Towne view. This study included the W4D-CTA images of acute stroke patients with occlusion of the internal carotid artery or the middle cerebral artery (MCA) from April to July 2021. Images produced by 320r-ADCT were transferred to a workstation. Unedited AP-view images were automatically generated. Towne-view images were cut out for this study. Collateral status was evaluated as poor, intermediate, or good based on the visualization of the MCA peripheral branches. In addition, the IRR was assessed using intraclass correlation coefficients (ICC) (2,1). Fifteen patients were analyzed. In the unedited AP-view and cut-out Towne-view W4D-CTA images, the ICC (2,1) were 0.147 and 0.796, respectively. Cut-out Towne-view W4D-CTA images with substantial IRR are superior to the unedited AP-view images for assessing the anterior intracranial collateral status.

## 1. Introduction

Whole-brain four-dimensional computed tomography angiography (W4D-CTA) using 320-row area detector computed tomography (320r-ADCT) can visualize flow dynamics in all intracranial arteries. Therefore, W4D-CTA has been applied to the diagnosis of intracranial artery occlusion and collateral status evaluation for mechanical thrombectomy [1,2,3,4,5,6]. Advances in the 2020-year model 320r-ADCT with a high processing power can shorten reconstruction time compared with the 2010-year model 320r-ADCT [4], and advances in workstations can automatically generate the subtracted 4D-CTA images. Endovascular physicians require images with a high degree of agreement among independent observers for making appropriate decisions regarding the thrombectomy process. However, the 320r-ADCT gantry cannot be tilted in volume scanning (VS); therefore, the head position influences the original W4D-CTA images generated in a workstation. When the original anteroposterior (AP)-view W4D-CTA images are transferred to picture archiving and communication systems (PACS) without any editing, W4D-CTA images can reach endovascular physicians quickly. However, in digital subtraction neuroangiography, Towne-view projections, defined as an AP view with a cephalocaudal angulation, are standard for visualizing anterior intracranial arteries [7]. Therefore, Towne-view W4D-CTA images may be more appropriate for the collateral status evaluation of anterior intracranial arteries before mechanical thrombectomy than the unedited AP-view images. However, it is uncertain which W4D-CTA images are better. Furthermore, a high level of patient radiation exposure in a 320r-ADCT compared to a 64-row CT may be problematic [8]. This retrospective study aimed to measure the radiation dose of W4D-CTA using the 2020-year model 320r-ADCT and determine which W4D-CTA images using the 2020-year model of 320r-ADCT are more appropriate pre-thrombectomy, whether the unedited AP view or the cut-out Towne view.

## 2. Materials and Methods

For this retrospective study, we searched for patients in the Institutional Electronic Medical Record and included patients who: were admitted between April 2021 and July 2021 due to acute ischemic stroke; underwent mechanical thrombectomy (MT) for occlusion of the internal carotid artery (ICA), the M1, or the proximal M2 segment of the middle cerebral artery (MCA); underwent W4D-CTA using the 2020-year model 320r-ADCT (Aquilion One PRISM Edition, Canon Medical Systems, Otawara, Tochigi, Japan) with a high processing power upon arrival. We excluded patients who underwent 80-row ADCT before mechanical thrombectomy [9], did not undergo mechanical thrombectomy, or did not have occlusion of the ICA or MCA.

### 2.1. Volume Scanning by the 320r-ADCT

Postcontrast VS was performed by injecting 40 mL of a non-ionic contrast medium (iopamidol; 370 mg/mL) at 4 mL/s. A 320 × 0.5 mm detector row CT system was used; however, VS was performed using 1 mm thickness with a z-axis coverage of 16 cm and 1 s single rotation intermittent dataset scans at 80 kV to shorten the image processing time and reduce radiation doses. The mask was acquired at 1 s and beginning 10 s after injection of contrast medium; 24 intermittent VSs were acquired at 80 mA. A total of 25 volumes of data were programmed for acquisition. The total computed tomography dose index (CTDI) volume was designed to be 96.6 mGy, and the total dose length product (DLP) was 1545 mGy∙cm. Each volume comprised 160 images, and 4000 images were programmed for production. Volume data were automatically transferred without subtraction to a workstation (Vitrea version 7.8, Canon Medical Systems). It took 268 s (4 min and 28 s) for 4000 images to reach the workstation after the acquisition was started. Brain perfusion application (CT Brain Perfusion 4D) was selected, and unedited, original, AP-view W4D-CTA images were automatically generated by subtraction (Figure 1 and Figure 2 Left) and transferred to PACS.

For this study, radiographers manually omitted the venous sinuses, cut out intracranial arteries, generated Towne-view 4D-CTA images (Figure 2 (left) and Figure 3) in the workstation (Appendix A), and transferred them to PACS. The time required to operate the workstation was approximately 20 s.

### 2.2. Diagnosis of Occlusion

Mechanical thrombectomy was determined based on neurological deficits, unedited W4D-CTA findings, and findings of CT perfusion using the Vitrea [10]. Digital subtraction angiography for mechanical thrombectomy demonstrated occlusion of the ICA, the M1 MCA, or the proximal M2 MCA.

### 2.3. Evaluation

The number of VS, total CTDI volume, and total DLP were evaluated. In addition, four raters (TM, TS, HS, and NM) independently evaluated the collateral status of the artery as poor, intermediate, or good, based on the visualization of the MCA peripheral branches in both 4D-CTA images [9] provided by the PACS.

### 2.4. Statistical Analysis

Non-normally distributed continuous variables were expressed as medians and interquartile ranges. Interrater reliability of collateral status evaluation among the four raters was assessed using intraclass correlation coefficients (ICC) (2,1), and the interrater reliability of the unedited and edited views was analyzed. Interrater reliability was defined as values < 0.10 indicating “no agreement”, ≥0.10 and <0.20 as “slight”, ≥0.20 and <0.40 as “fair”, ≥0.40 and <0.60 as “moderate”, ≥0.60 and <0.80 as “substantial”, and ≥0.80 and ≤1.0 as “almost perfect agreement”. We used the JMP software (version 16.2; SAS Institute, Cary, NC, USA) for all statistical analyses. One author (TM) had full access to all study data and took responsibility for the integrity and analysis of the data.

## 3. Results

Among the 119 patients with acute ischemic stroke, 15 of them met our inclusion criteria (Appendix A). Five patients had ICA occlusion, and ten patients had MCA occlusion. The median number of VSs was 25, the median total CTDI volume was 96.6 mGy, and the median total DLP was 1545 mGy∙cm (Appendix A). Four raters independently assessed the collateral status of the unedited AP view and the cut-out view W4D-CTA images (Appendix A). Only two cases (Nos. 6 and 15) in the unedited AP-view 4D-CTA received the same evaluation from the four raters compared to the corresponding nine cases (Nos. 4, 5, 7, 8, 10, 11, 12, 13, and 15) in the Cut-out Towne-view 4D-CTA (Appendix A). The ICC (2,1) values were 0.147 and 0.796, respectively. Cut-out Towne-view of the W4D-CTA images (Appendix A) had a higher degree of agreement among the four raters than the unedited AP-view W4D-CTA images (Appendix A).

## 4. Discussion

Our results demonstrated that radiation exposure of W4D-CTA using the 2020-model 320r-ADCT was not high, and cut-out Towne-view W4D-CTA images were superior to the unedited AP-view images because the former had a higher degree of agreement among independent raters. Therefore, cut-out Towne-view images can help endovascular physicians to make appropriate decisions for thrombectomy of occlusion of the ICA or the MCA.

Radiation dose reduction is crucial in a multidetector CT. The DLP of intracranial CTA and CT perfusion using a 64-row CT scanner was 386 mGy∙cm and 1037 to 2330 mGy∙cm, respectively, although the scan length of CT perfusion using a 64-row CT scanner was 24 mm only [11]. The total DLP of 4D-CTA using an 80-row ADCT with 4 cm coverage was 690 mGy∙cm/25 scans [12]. In our study, technological advancements in the 2020-model 320r-ADCT, such as low tube current, low peak voltage, collimation, and deep learning reconstruction, contributed to dose reduction [13], and the median total DLP was 1545 mGy∙cm/25 scans. The z-axis coverage in the 320r-ADCT is 16 cm, i.e., 4 times; however, the total DLP was 2.2 times. For optimization of maximum radiation dose, the peak voltage appropriate to the patient, the tube current appropriate to the patient, noise reduction mechanism, and the appropriate z-axis coverage are important [14]. Deep learning reconstruction lowered image noise and radiation exposure compared to conventional hepatic dynamic CT [13,15].

In many stroke centers, magnetic resonance angiography or three-dimensional (3D) CTA is used to find occlusion of the intracranial arteries, and magnetic resonance perfusion or CT perfusion is essential to evaluate penumbra and ischemic core in the ischemic area before mechanical thrombectomy [16,17,18]. However, magnetic resonance angiography or 3D-CTA cannot visualize flow dynamics in intracranial arteries. Furthermore, when 3D-CTA is used to find occlusions of intracranial arteries, CT perfusion must be added and total radiation exposure of 3D-CTA and CT perfusion increases. In contrast, whole-brain VS can perform W4D-CTA and CT perfusion simultaneously, and total radiation exposure can be reduced. Therefore, W4D-CTA, which can visually provide information on flow dynamics such as digital subtraction neuroangiography, is a promising tool to find occlusion of intracranial arteries and detect the ischemic area in an acute stroke setting [1,3,5].

Dynamic axial computed tomographic angiography using an 80-row ADCT can visualize the ICA or MCA occlusion. The acquisition began by angulating the gantry with a narrow width to focus on the MCA axis [12]. However, the 320r-ADCT cannot be tilted in VS, the acquisition cannot focus on the MCA axis in VS, and the patient’s head position influenced the quality of the unedited W4D-CTA images. Therefore, radiographers must decide whether the unedited W4D-CTA images are appropriate for collateral status evaluation. If the unedited W4D-CTA images are inappropriate, radiographers should edit them for endovascular physicians to make appropriate decisions and provide edited images to endovascular physicians.

There are a few studies reporting classification of dynamic CTA. A six-point ordinal scale was reported for multiphase CTA [19] and a five-point ordinal scale for W4D-CTA [3,20]. These previous studies classified CTA images into three and two categories of collateral status, respectively. In this study, we used three categories of good, intermediate, and poor collateral status [9]. If W4D-CTA is widely used for decision making in acute ischemic stroke, the standardization of categories of collateral status is necessary. Standardization is the process of developing and agreeing upon criteria or methods that can increase repeatability and quality [21].

Few studies have reported the application of the 2010-year model 320r-ADCT for acute ischemic stroke because of the relatively long image reconstruction and subtraction time of the volumetric data [2,4]. However, the 2020-year model 320r-ADCT and workstation technological advances shortened the image processing time. When brain perfusion application (CT Brain Perfusion 4D) of the Vitrea was selected, not only W4D-CTA but also CT perfusion maps were automatically generated, which demonstrated the ischemic area and penumbra [10,19]. Several studies have reported on the demonstration of the ischemic core and penumbra in workstations [22,23].

We used the ICC (2,1) to assess the absolute agreement among raters, because the ICC (2,1) is used when each subject is measured by each rater, and raters are considered representative of a larger population of similar raters. Reliability is calculated from a single measurement [24,25,26]. Furthermore, the minimum sample size is 14 for calculating three rater’s ICC (2,1), when the expected absolute agreement is 0.8, and the average length of the confidence interval is bounded by 0.4 [27]. Therefore, our sample size of 15 was enough, when the expected absolute agreement was 0.8, and the average length of the confidence interval was bounded by 0.4. If a target agreement decreases, the required sample size increases. If the average length of the confidence interval decreases, the required sample size increases [27]. In our study, the ICC (2,1) for cut-out Towne-view W4D-CTA images was 0.796, and the sample size of 15 was enough for assessing the absolute agreement among four independent raters.

Radiation exposure of the latest W4D-CTA is reduced compared to that of a combination of 3D-CTA and CT perfusion using a 64-row CT. Cut-out Towne-view W4D-CTA images had a higher degree of absolute agreement among independent raters than the unedited AP-view W4D-CTA. Therefore, those images in combination with CT perfusion may be essential in identifying candidates for mechanical thrombectomy [20]. If W4D-CTA is widely used for decision making in acute ischemic stroke, the standardization of the W4D-CTA protocol such as contrast medium volume, concentration of contrast medium, rate of contrast medium injection, collimation, peak voltage, tube current, z-axis coverage, and DLP is required.

### Limitations

Our study has several limitations; first, it was based on a retrospective design. Therefore, selection bias might have occurred during the selection of the study participants among acute stroke patients in our institution. Our study population is not representative of acute stroke patients who underwent mechanical thrombectomy. Misclassification of the collateral status evaluation might have introduced information bias [28,29]. Second, different views of W4D-CTA images may have a higher degree of agreement than that of the cut-out Towne-view images because the other views of the W4D-CTA images aside from the cut-out Towne-view or unedited AP-view images were not evaluated. Third, there may be a better category for classifying CTA images than our three categories. Fourth, the study dealt with 320r-ADCT only, and the generalizability of the study results to different rows of ADCTs is uncertain. Fifth, our study was conducted using images and volume data of patients who underwent mechanical thrombectomy. Therefore, it is unknown whether the cut-out Towne-view W4D-CTA images can identify candidates more appropriately than the unedited AP-view W4D-CTA. Finally, we did not assess whether the cut-out Towne-view W4D-CTA images were beneficial for favorable clinical outcomes after mechanical thrombectomy. Therefore, a prospective study is warranted to establish whether cut-out Towne-view W4D-CTA images can contribute to a favorable clinical outcome after mechanical thrombectomy in a larger sample size.

## 5. Conclusions

Radiation exposure of the latest W4D-CTA is reduced in spite of 16 cm z-axis coverage compared to that of a combination of 3D-CTA and CT perfusion with 2.4 cm z-axis coverage. Cut-out Towne-view W4D-CTA images by the 320r-ADCT with a high processing power had a high degree of agreement among raters for assessing the collateral status of the MCA or ICA occlusion. A prospective study is warranted to establish whether endovascular physicians can make the same decisions using those images and whether cut-out Towne-view W4D-CTA images can contribute to the achievement of a favorable clinical outcome after mechanical thrombectomy.

## Figures and Tables

**Figure 1 diagnostics-12-01336-f001:**
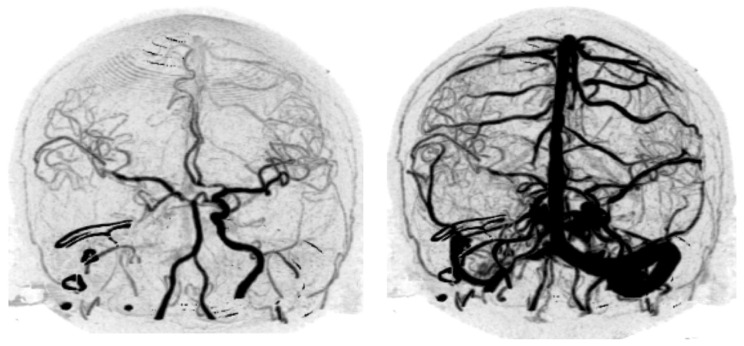
Unedited anteroposterior-view W4D-CTA images of Case 8. (**Left**) Arterial phase; (**Right**) Arterial and venous phase. W4D-CTA, whole-brain four-dimensional computed tomography angiography.

**Figure 2 diagnostics-12-01336-f002:**
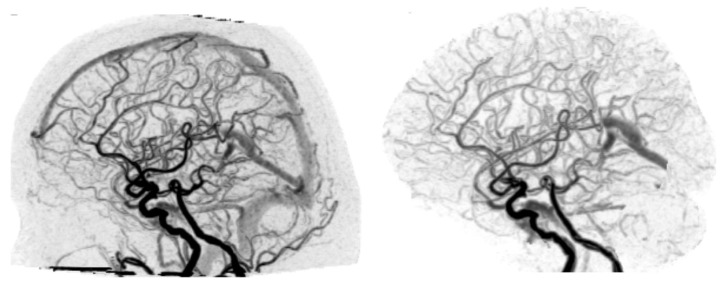
(**Left**) Unedited lateral-view W4D-CTA image of Case 8; (**Right**) Edited lateral-view W4D-CTA image of Case 8 where venous sinuses were omitted. W4D-CTA, whole-brain four-dimensional computed tomography angiography.

**Figure 3 diagnostics-12-01336-f003:**
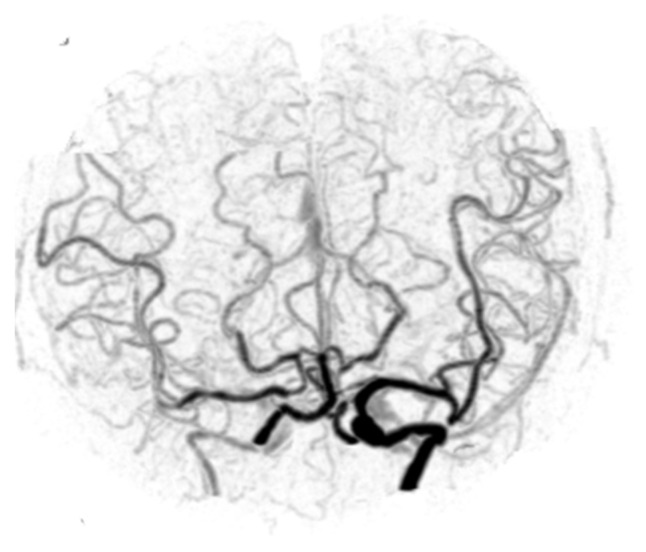
Cut-out Towne-view W4D-CTA image of Case 8. W4D-CTA, whole-brain four-dimensional computed tomography angiography.

## Data Availability

All data generated or analyzed during this study are included in this published article.

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
