# Peer review of "Cut-Out Towne-View Whole-Brain 320-Row Four-Dimensional Computed Tomography Angiography for Assessing the Anterior Intracranial Collateral Status: A Retrospective Study"

_diagnostics, 2022, doi:10.3390/diagnostics12061336_

Round 1

Reviewer 1 Report

The clinical problem is well approached.

The justifications of the method is appropriate.

I have no particular suggestions to propose.

Author Response

Thank you for the time and effort you expended in reviewing our manuscript and providing comments.

Reviewer 2 Report

This is a very interesting and clinically important paper. Despite some minor limitations such as relatively small study group and retrospective character the methodology and soundness of the results are very interesting.

May authors describe in more details:

1. The methodology of manual cut out: (Line 132, 133: radiographers manually omitted the venous sinuses, cut out intracra- 132 nial arteries, generated Towne-view 4D-CTA images). Maybe a sceen shot from workstation would be a good idea showing that most important step for the procedure.

2. Results - may authors present them in a more readible way? The Table 2 is very difficult to analyse and see the superiority of the new method in comparison to non-edited images. Maybe some graphical form would be valuable to be added.

It would be very valuable if authors would perform prospective study on larger population and also perform intra-rater and inter-rater analyses.

Author Response

Response: Thank you for the time and effort in reviewing our manuscript and providing comments and suggestions, which have considerably helped us improve our manuscript. We have provided point-by-point responses to your comments below and made revisions per your suggestions.

  1. The methodology of manual cut out: (Line 132, 133: radiographers manually omitted the venous sinuses, cut out intracranial arteries, generated Towne-view 4D- CTA images). Maybe a sceen shot from workstation would be a good idea showing that most important step for the procedure.

Response: We have captured a screenshot of cutting out intracranial arteries in the Vitrea workstation. We have provided the image as Figure S1 in the supplementary file.

Figure S1. A screenshot of cutting out intracranial arteries (B: red line) in the Vitrea workstation

  1. Results - may authors present them in a more readible way? The Table 2 is very difficult to analyse and see the superiority of the new method in comparison to non-edited images. Maybe some graphical form would be valuable to be added.

Response: The four raters’ evaluation of good, intermediate, or poor is described in Table S2 (Supplementary file). The evaluation cannot be converted to a graphical form. However, we have added a few sentences in the Results, page 4, line 164, to explain the visual difference between the unedited AP-view and Cut-out Town-view 4D-CTA, as follows: “Only two cases (no.6 and 15) in the unedited AP-view 4D-CTA got the same evaluation from the four raters compared to corresponding nine cases (no.4, 5, 7, 8, 10, 11, 12, 13, and 15) in the Cut-out Town-view 4D-CTA (Supplementary file, Table S2).”  

Reviewer 3 Report

The paper provides a comparison between W4D-CTA (16 cm z-axis coverage) and the combination of 3D-CAT plus CT perfusion (2.4 cm z-axis coverage), respectively, in what it concerns radiation exposures. Also, it is shown that the cut-out Towne-view images are able to provide the necessary information for assessing the collateral status of MCA or ICA occlusions. As pointed by the authors, it should demonstrate also, in a further stage of this work, if this kind of images lead to same physician decision. In my opinion this is not trivial as physicians experience has been developed based on a different kind of images.  

The paper is well written, providing the necessary details and good references. Therefore, I recommend its publication in Diagnostics.

Minor remarks: Fig. 4 is not really necessary, Tables 1-2 could be relocated to the supplementary section.

Author Response

Response: Thank you for the time and effort in reviewing our manuscript and providing comments and suggestions, which have considerably helped us improve our manuscript. We have provided point-by-point responses to your comments below and made revisions per your suggestions.

  • Minor remarks: Fig. 4 is not really necessary, Tables 1-2 could be relocated to the supplementary section.

Response: We have relocated Figure 4 and Tables 1 and 2 to the supplementary section as Figure S2 and Tables S1 and S2, respectively.